# Emulgels: Promising Carrier Systems for Food Ingredients and Drugs

**DOI:** 10.3390/polym15102302

**Published:** 2023-05-13

**Authors:** Jovana Milutinov, Veljko Krstonošić, Dejan Ćirin, Nebojša Pavlović

**Affiliations:** Department of Pharmacy, Faculty of Medicine, University of Novi Sad, Hajduk Veljkova 3, 21000 Novi Sad, Serbia

**Keywords:** emulsion gels, delivery systems, drug delivery, topical formulations

## Abstract

Novel delivery systems for cosmetics, drugs, and food ingredients are of great scientific and industrial interest due to their ability to incorporate and protect active substances, thus improving their selectivity, bioavailability, and efficacy. Emulgels are emerging carrier systems that represent a mixture of emulsion and gel, which are particularly significant for the delivery of hydrophobic substances. However, the proper selection of main constituents determines the stability and efficacy of emulgels. Emulgels are dual-controlled release systems, where the oil phase is utilized as a carrier for hydrophobic substances and it determines the occlusive and sensory properties of the product. The emulsifiers are used to promote emulsification during production and to ensure emulsion stability. The choice of emulsifying agents is based on their capacity to emulsify, their toxicity, and their route of administration. Generally, gelling agents are used to increase the consistency of formulation and improve sensory properties by making these systems thixotropic. The gelling agents also impact the release of active substances from the formulation and stability of the system. Therefore, the aim of this review is to gain new insights into emulgel formulations, including the components selection, methods of preparation, and characterization, which are based on recent advances in research studies.

## 1. Introduction

In recent years, drug carriers have been of great scientific and industrial interest, which has led to significant developments and profits. They are defined as systems that have the ability to incorporate and protect a drug to improve its selectivity, bioavailability, and efficacy [1]. The development of drug delivery systems in order to achieve the therapeutic effects of drugs has contributed to the treatment of various diseases and has led to scientific progress in the field of drug therapy [2,3]. The use of drug carriers is considered a way to overcome problems such as low bioavailability, poor solubility in water, and biological degradation of most medicinal substances [4].

Topical drug delivery systems are commonly used to achieve localized effects on the skin for both cosmetic and dermatological purposes for application on healthy or diseased skin. They are generally employed for the treatment of skin disorders such as eczema, acne, and psoriasis. However, locally applied formulations may exert systemic effects through their transdermal delivery. This administration of drugs has many benefits such as avoiding first-pass metabolism, avoiding degradation in gastric and frequent dosing, improving patient compliance, and easy self-medication. These systems include solid, semi-solid, and liquid formulations such as powders, creams, ointments, lotions, emulsions, etc. [5,6].

Hydrogels are modern, promising, and smart drug delivery vehicles due to their extremely tunable physical characteristics that ensure the minimization of the drawbacks of conventional drug delivery. They are usually made of up to 90% of water; they necessarily contain a polymer that forms a three-dimensional network structure by cross-linking its chains, which makes hydrogels porous to accommodate drugs. The main disadvantage of hydrogels is the impossibility of delivering hydrophobic drugs, which represents a major challenge because most effective drug substances are hydrophobic [1,7]. In order to overcome this disadvantage, emulgels are developed that are also known as creamed gels and gelled emulsions. Emulgels are novel drug delivery systems prepared by mixing emulsions and gels; therefore, they possess properties of both emulsions and gels. Consequently, emulgels are dual-controlled release systems with numerous preferable characteristics as well as high patient acceptability [8,9]. Emulgels possess numerous favorable properties such as being easily spreadable, easily removable, greaseless, and having a pleasing and transparent appearance [10]. Compared to other topical and transdermal formulations such as ointments, solutions, and patches, emulsion-based formulations possess better chemical, physical, and biological compatibility with the skin. For instance, ointments block the normal evaporation of moisture from the surface of the skin and are greasier, which leads to discomfort. Compared to solutions, the viscosity of emulsions can be easily adjusted to prevent excessive spreading on the skin after application. Emulsion-based formulations are compatible with both hydrophilic and lipophilic drugs. Therefore, emulsions are the most frequently used formulations in dermatology and cosmetology. Emulsions are not simple systems, on the contrary, they are heterogeneous systems consisting of two immiscible liquid phases in which one phase is dispersed in the form of droplets in the other one. The dispersed phase represents a reservoir that carries the active substance [11,12].

Additionally, emulgels have recently emerged as promising delivery systems for hydrophilic as well as lipophilic functional ingredients and nutraceuticals such as carotenoids, vitamins, probiotics, and unsaturated fatty acids. Emulgels are used to overcome low chemical stability, low solubility, and the poor absorption of food ingredients, and to improve the sensorial textures of formulations, bioavailability, bioaccessibility, and control their release [13,14,15]. Generally, various food products can be categorized as emulgels, especially protein-based emulsions that can be easily converted into soft solids by heating, acidification, or enzyme action [16]. Therefore, protein-based emulsion gels have attracted the attention of researchers. The most commonly employed biopolymers are casein, soy, gelatin, and whey protein, due to their promising emulsifying and gelling properties as well as their abundance and renewable potential. Proteins can be used alone or in combination with polysaccharides (e.g., agar, gellan, xanthan gum) as excipients in the formation of food emulsion gels [13]. These systems have emerged as promising biomaterials with tunable properties and with the potential to be employed for the protection of health-promoting food ingredients, as well as to develop food products such as viscoelastic gels with zero trans fats or less saturated fats as a substitute for solid fats [17]. One of the advantages of food emulgels, in comparison to emulsions, is also the slower intestinal release of active substances due to improved gastric and intestinal stability [18]. This can be explained by the changes in the structures of emulsion gels during eating. Namely, it has been shown that the three-dimensional gel network largely inhibits the attachment of lipases onto the surface of oil droplets, which decreases lipid digestion and release of the incorporated ingredients [19]. In emulgels, oil droplets are less mobile and less accessible to the digestive enzymes, and the active ingredients within the oil droplets are typically more stable and have sustained release in the gastrointestinal tract [14]. Proper selection of emulgel excipients may thus provide the desired release profile in a controlled manner. In addition, emulgels are increasingly being investigated as stimuli-responsive systems which may alter their morphology and properties upon exposure to external stimuli such as enzymes, temperature, pressure, light, or microorganisms [13].

The aim of this review is to gain a broad overview of emulgels as a modern approach for the delivery of active substances, including formulation consideration, preparation, and characterization of emulgels based on recent advances in research studies on emulgels.

## 2. Formulation of Emulgel

The properties of emulgels such as non-toxic, non-irritating, non-sensitizing, and non-comedogenic are mainly associated with the emulgel composition. The primary requirement for the formulation of an emulgel is the appropriate selection of the oil phase, emulsifier, and gelling agent [5,20].

The proper selection of the emulgel excipients would lead to the adequate release of active ingredients, their permeation into/through the biological barriers (skin or intestinal mucosa), and consequently their biological or pharmacological effects.

Controlled release and targeted drug delivery can be achieved through the dual controlled release mechanisms, due to the presence of an emulsion and a gel system [8]. Besides the physiochemical characteristics of the active substances (molecular weight, solubility, partition coefficient, etc.) and the physiological factors (properties of biological barriers, such as membrane thickness, pH, blood flow, etc.) [21], the formulation factors were also shown to greatly impact the release and membrane transport of active substances. The particle size of a colloidal system (e.g., an emulsion in an emulgel) has a key role in affecting the release profile of substances incorporated inside the particles, with the smaller sizes increasing the amount of active substances released and penetrating the skin. Surfactants, which are used to disperse and emulsify one phase in another, affect the drug release through their effects on oil droplet size but also on the structure of the lipid particles [22]. Polymers used as gelling agents in emulgels enhance the physical stability of an emulsion by increasing the viscosity of the continuous phase, thus directly providing a slower release of active substances [23]. Furthermore, it should be noted that an increase in the drug loading in a formulation accelerates the release rates, as well as the presence of penetration enhancers in emulgels [11,24]. The factors affecting the release of active ingredients from emulgel formulation and their permeation are summarized and illustrated in Figure 1.

As was already stated, emulgels are the carrier systems that are particularly significant for the delivery of hydrophobic active ingredients. Specifically, emulsions with an interfacial layer of adsorbed globular proteins may be considered “soft capsules” [25]. Belonging to lipid-based drug delivery systems, they largely differ from polymeric and inorganic carrier systems. Lipid-based delivery systems are classified into vesicular carriers (liposomes, ethosomes, etc.) and lipid particulate systems [26]. Each class of drug carrier systems has numerous advantages and disadvantages regarding cargo, delivery, and patient response. Formulation simplicity, high loading capacity, biocompatibility, and high bioavailability of lipophilic ingredients are listed as the main advantages of lipid-based carriers, while polymeric carrier systems can be synthesized from natural or synthetic materials, allowing them tunable structures and properties [27]. Vesicular lipid systems have the main advantage of being able to carry both hydrophobic and hydrophilic drug molecules. While synthetic liposomes have multifunctional flexibility, natural biomembrane materials are increasingly studied due to their interesting biomimetic properties [28]. Outer membrane vesicles originating from bacteria are of particular importance. Polypeptide vesicles are also widely investigated since they are composed of hydrophobic and hydrophilic chains and they can encapsulate hydrophobic drugs through hydrophobic interactions, hydrophilic drugs through electrostatic interactions, and metal-based drugs via metallic coordination [29]. Drug release from a polymeric carrier is impacted by its composition, ratios of drug, polymer and excipient, interactions between constituents, and the methods of preparation, and it can be exerted via four different mechanisms (diffusion, solvent, chemical interaction, and stimulated release), which makes them suitable for targeted drug delivery [30]. Nanoporous polymers, such as those forming porous hollow polymeric capsules, are being increasingly investigated for various biological applications due to their high capacity for incorporating drug substances. Carbonyl-functionalized hollow organic capsules are particularly significant since some biomolecules, such as folic acid, can be covalently bound on them and efficiently delivered [31].

### 2.1. Gelling Agents

Gelling agents are dissolved or dispersed in a suitable medium and provide a weakly cohesive three-dimensional structural network with a high degree of cross-linking, either physically or chemically, to produce semisolid systems. These agents are chiefly used at a concentration of 0.5–10% in semisolid formulation [32,33].

Gelling agents used in emulgel formulation are divided into natural, synthetic, and semi-synthetic based on their origin. Natural gelling agents ensure remarkable biocompatibility and biodegradability, but on the other hand, their major drawback is microbial degradation. Bio-polysaccharides or their derivatives and proteins belong to natural gelling agents. For instance, bio-polysaccharides are pectin, carrageenan, alginic acid, and gelatin, whilst xanthan gum, starch, and dextran are derivatives of bio-polysaccharides [34]. However, various kinds of both semi-synthetic and synthetic gelling agents are widely used in emulgel formulation [5]. Semi-synthetic gelling agents are better than natural ones regarding their stability and resistance to environmental influences [34]. Cellulose derivatives are semi-synthetic agents that are extensively employed in topical formulation such as methylcellulose, carboxymethylcellulose, and hydroxypropylmethylcellulose (HPMC). The major disadvantages of cellulose derivatives are that they are incompatible with various drugs and require the use of preservatives in their formulation in order to prolong their storage period [35]. Nowadays, carbomer as a synthetic polymer is widely used as a gelling agent in semisolid formulations and represents a macromolecular polymer of acrylic acid [32]. Carbopol^®^ 940 is a commercially available acrylic acid polymer that is often used as a gelling agent in the pharmaceutical industry due to its hydrophilic, non-toxic, and non-irritating properties, as well as its ability to form gels over a wide pH range at room temperature [36].

Natural gelling agents such as soy protein, whey protein, pectin, carrageenan, and others are commonly used in emulsion gels for the delivery of functional food ingredients [14]. Soy protein isolate (SPI) is a good alternative to animal-based protein due to its low-cost, good nutritional values, and superior functional properties. However, soy protein isolate does not form a gel with good stability properties, and polysaccharides are commonly added to improve gel properties. Xiao et al. have proven that wheat bran cellulose effectively improves the gel properties of SPI and enables the development of formulation with desirable functional properties [37]. Generally, protein-based emulsion gels have been prepared by special gelation techniques, both heat-set and cold-set, with approaches such as acidification, enzyme treatment, ethanol-induced, salt addition, and hydrostatic pressure-induced [13].

Recently, alginate-based food emulgels have received great attention and they are formed by ionic cross-linking of alginates with divalent cations, mostly calcium. Alginates were used in combination with both conventional [38] and protein emulsifiers [39,40] in the formulation of emulgels. It was shown that the addition of sodium alginate in the systems emulsified by Tween 80 improves the gel stiffness after freeze-thawing and that alginate-stabilized emulsion gels might be used in the preparation of low-fat mayonnaise products and other similar emulsion foods [38]. For the emulsions containing soy protein isolate and sodium alginate, it was demonstrated that the concentration of alginate, as well as concentrations of protein and oil phase, impact the gelation process and formation of emulsion gel beads, which may further affect encapsulation, stability, and release of the encapsulated hydrophobic active substances [39]. Similarly, the emulsion-gelled microparticles were prepared from whey protein isolate and sodium alginate as well, in order to be used as carriers of hydrophobic nutrients and bioactive substances [40].

Table 1 lists some gelling agents and their concentrations used in the preparation of emulgel formulations.

Gelling agents are markedly important components of the emulgel employed to improve dosage form quality [5]. The incorporation of gelling agents into emulsions leads to the formation of a gelled structure, which makes the system thixotropic [59]. Thixotropy refers to the phenomenon of fluid to reversible structural transition, i.e., gel–sol–gel conversion due to the time-dependent changes in the viscosity under certain conditions. Gel–sol–gel behavior imparts stability and increases the bioavailability of the system [60].

The gelling agents impact the consistency of dosage form, spreading coefficient, viscosity of the formulation, drug release from the formulation, and stability of a system. Generally, gelling agents are used to increase the consistency of any formulation [61].

Spreadability is one of the most important properties of topical formulations that affect therapeutic efficacy and patient compliance. Nikumbh et al. have shown that the spreading coefficient depends on both the type and concentration of gelling agents employed in the formulation [56,62].

Viscosity is an important physical property for topical drug delivery systems, which usually indicates the consistency of any semisolid formulations. Gelling agents extremely affect the spreadability and viscosity of the formulation, and consequently impact the drug release from the formulation [49,63]. Daood et al. have proven that the viscosity of the emulgels increases as the concentration of gelling agents increases. In addition to the concentration, they have shown that the viscosity is also affected by the type of gelling agent [64]. Shahin et al. have observed that the extent of release is affected by the amount of gelling agent, with the inverse correlation between them. In order for the drug to be released from the carrier, which in this case is an emulsion, and reach the biological membrane, transport within the formulation occurs either by molecular diffusion or by diffusion and convection of the oil droplets. By increasing the viscosity of the formulation, which occurs when the concentration of the gelling agent increases, the diffusion path increases and, consequently, the release rate decreases [65].

Except for all the aforementioned, gelling agents are used to stabilize topical formulations [34]. The influence of the gelling agent on the stability of the preparation was presented in a study by Abdel-Bary et al., where it was shown that the combination of two types of gelling agent leads to an increase in stability, i.e., synergism occurs. They found that a chloramphenicol-based gel containing a mixture of Carbopol 940 and hydroxyethyl cellulose was physically more stable than those containing either Carbopol 940 or sodium carboxymethyl cellulose alone [66].

### 2.2. Oil Phase

The oil phase is utilized as a carrier for hydrophobic drugs and as well as it determines the physical properties of the final product such as occlusive and sensory properties. In addition, the oil phase can influence the viscosity, permeability, and stability of the emulsion and also drug release. For pharmaceuticals, cosmetics, and the food industry, lipids of both natural and synthetic origin are generally employed. For external application, mineral oils either alone or in combination with soft or hard paraffin are widely used, while vegetable oils are the most commonly used in oral preparations [5,20,67,68]. The oil phase has an influence on the pharmacological effects of pharmaceutical formulations, drug release from the formulations, spreadability, and the occlusive effect of the formulations.

Depending on the selected oil phase, the oil may give a synergistic effect with active substances as various oils have medicinal benefits [20,68]. Some vegetable oils have a sun protection effect that has been proven in numerous studies, for example, sesame, coconut, peanut, olive oil, and cottonseed oil [69]. Montenegro et al. have examined the usage of vegetable oils to improve the sun protection effect of topical sunscreen formulations and they have proven that the incorporation of pomegranate and shea oil into formulations containing different percentages of organic UV filters improved the SPF values [70]. In another study, ginger, and garlic oils were proven to enhance wound healing [71].

The existence of the influence of the oily phase on the release of the drug from the emulgel was proven in the study by Khullar et al., who examined emulgel formulations of mefenamic acid and demonstrated that a higher rate of drug release was achieved in formulations with a lower proportion of the oily phase. The same study showed that the proportion of oil phase in emulgel formulations had the same effect on spreadability as the proportion of gelling agent but was less pronounced [72].

It is known that the presence of an oil phase in the formulation leads to occlusive effects on the skin. The type and amount of the oil phase, as well as the type of emulsifier, affect the value of the occlusive factor, which was shown in the study by Montenegro et al. [73].

### 2.3. Aqueous Phase

Aqueous materials that are the most commonly used to prepare aqueous phase are water and alcohol [74].

### 2.4. Emulsifiers

The emulsifiers are used to promote emulsification during the preparation process and to ensure the physical stability of the emulsion during its shelf life. The choice of emulsifying agents is based on their capacity to emulsify, their toxicity, and the route of administration. Nonionic surfactants are chiefly used in the formulation for biomedical applications due to their low toxicity and compatibility with most active substances. The most commonly used emulsifiers in emulgels are Tween 20 (polysorbate 20) as an emulsifier in the aqueous phase, and Span 20 (sorbitan monolaurate 20) in the oil phase. Other examples of emulsifiers used in emulgel formulations are sorbitan monooleate, polyethylene glycol stearate, stearic acid, and sodium stearate [29,67,75].

Proteins possess amphiphilic, film-forming properties, and biocompatibility and they are thus used as emulsifiers, primarily in the food industry. Proteins stabilize emulsions by being adsorbed on the oil–water interface leading to reduced interfacial tension and viscoelastic interface film as well as previously mentioned surfactants [76]. In addition, protein aggregates are usually insoluble, thus forming Pickering emulsions and preventing droplet flocculation and/or coalescence through the electrostatic repulsion and steric hindrance between protein-coated emulsion droplets as well [77]. The emulsifying capacity of proteins largely depends on protein sources, structure, molecular weight, and adsorption behavior. Traditionally, the most commonly used protein-based emulsifiers in foods are dairy proteins (casein and whey protein). However, plant proteins have been increasingly used as alternatives, and the most widely investigated plant proteins in food formulations include soy proteins, pea proteins, lupin, cowpea, wheat gluten, rice glutelin, etc. [78]. Soy protein is widely available on the market and exhibits considerable potential to be used in different formulations as an emulsifier more than as a gelling agent due to its characteristics [79].

Interfacial rheology of protein-stabilized emulsions is of utmost importance as well. Proteins that adsorb at the oil/water interface can be either compact and globular or flexible and random coil. Both globular and flexible proteins form the interfacial films, which can be considered as a thin interfacial gel layer [25]. Globular proteins, such as β-lactoglobulin, upon adsorption on the interface tend to have a definite orientation, with their hydrophilic groups remaining in the aqueous phase and the hydrophobic groups facing toward the oil phase. Therefore, the interfacial layer formed by globular proteins is more compact and resistant to deformation, which results in high viscoelasticity [80]. On the other hand, flexible proteins, such as β-casein, undergo rapid structural rearrangements at the interfaces, forming a train-loop-tail conformation in the adsorbed layer (Figure 2). Trains are the protein segments consisting mostly of hydrophobic groups that are strongly adsorbed at the interface, loops are the segments located between two points of contact with the interface, and tails are the segments between the interface contact point and the free end of the protein chain [81].

Using solid particles instead of surfactants as emulsifiers, Pickering emulsions and emulsion gels with specific characteristics may be formulated as well. Lu et al. utilized amphoteric lignin as a polymer for the in situ hydrophobization of silica nanoparticles via electrostatic adsorption, which resulted in the formation of pH-responsive oil-in-water Pickering emulsions with gel-like flow behavior [82].

Table 2 lists some emulsifiers and their concentrations used in the preparation of emulgel formulations.

### 2.5. Penetration Enhancers

Penetration enhancers are agents that increase drug penetration through the skin. These agents facilitate the absorption of drugs through various mechanisms such as temporarily disrupting the skin barrier, altering the partitioning of the drug into skin structures, fluidization of lipid channels between corneocytes, etc. Penetration enhancers commonly used in emulgels for topical drug delivery are oleic acid, clove oil, and menthol (Table 3) [91].

Properties of penetration enhancers [92]:These agents should be non-toxic, non-irritating, and non-allergenic;They should be pharmacologically inactive. Consequently, they should not bind to receptors;They should have predictable and reproducible activity and duration of effect;They should allow drugs into the body while preventing the loss of endogenous material from the body;They should be compatible with both excipients and active ingredients;They should possess cosmetic acceptability and be appropriate for the skin.

**Table 3 polymers-15-02302-t003:** List of penetration enhancers used in emulgel formulations.

Penetration Enhancers	Concentration Used (%w/w)	Reference
Menthol	1.0, 5.0, and 9.0	[93]
Clove oil	8.0 and 10.0	[72]
Mentha oil	4.0 and 6.0	
Oleic acid	7.7 and 7.8	[36]
Eucalyptus Oil	1.0, 3.0, and 5.0	[92]
Transcutol	1.0, 3.0, and 5.0	

### 2.6. pH Adjustment

The pH value of the emulgel formulation, generally of all topical formulations, should be compatible with skin pH, thereby avoiding the risk of skin irritation during application [94]. The pH of the adult skin is in the range of 4.1 to 5.8 and topical skin care preparations are adjusted to pH 5.4 or 5.5 due to preserving the “physiological” skin pH but can have a different pH value depending on their purpose [95]. TEA (triethanolamine) is commonly used for pH adjustment in emulgel formulations considering Carbopol is widely used as a thickener and among other things. TEA is necessary to uncoil the polymer chain of Carbopol and for the formation of a gel structure [65].

### 2.7. Preservatives

These are agents that protect the formulation from spoiling due to stopping or slowing microbial development [67]. The ideal preservative should be active against most microorganisms at a low concentration but this is not always the case. Parabens are more effective against fungi than bacteria, and antibacterial activity is less against Gram-negative bacteria than against Gram-positive organisms [96]. On the other hand, phenoxyethanol possesses a broad spectrum of antimicrobial activity and is effective against Gram-positive and Gram-negative bacteria and yeasts [97]. However, Schmitt et al. state that phenoxyethanol has low efficiency against fungi [98]. Examples of the most-employed preservatives are methylparaben, propylparaben, benzalkonium chloride, benzoic acid, and benzyl alcohol [67]. Some gelling agents are incompatible with certain preservatives, for instance, methylcellulose is incompatible with methylparaben, propylparaben, butylparaben, and cetylpyridinium chloride. Hence, we must take care to choose suitable preservatives [35]. In addition, the concentration of emulsifiers may influence the effectiveness of preservatives which was shown in the study of Puschmann et al. They found that an increased emulsifier concentration in emulgel formulations leads to decreased antimicrobial efficacy of phenoxyethanol due to the interaction between emulsifier and phenoxyethanol (more phenoxyethanol is solubilized in micelles) and a reduced free preservative concentration in the aqueous phase that is necessary for interactions with microorganisms [99]. Table 4 lists some preservatives and their concentrations used in the preparation of emulgel formulations.

### 2.8. Humectants

Humectants are often used in emulgel formulations to prevent loss of moisture and improve the characteristics of the formulation such as ease of application and consistency. Humectants mostly used in the topical formulation are glycerol and propylene glycol [94].

## 3. Preparation Processes of Emulgels

There are several preparation methods for emulsion gels. Generally, preparing emulgel is simple and involves three main steps (Figure 3). The emulsion and gel are prepared separately and mixed together at the end. Hence, the first step is the preparation of an emulsion, oil-in-water or water-in-oil, by preparing separately an oil and aqueous phase and mixing them together. This step includes heating the phases and incorporating the drug usually into an internal phase of the emulsion as required. Substances are added into a certain phase depending on their nature, such that hydrophobic substances are added into the oil phase and hydrophilic substances are added into the aqueous phase. After preparing the emulsion, the gel is made by adding a gelling agent to the water. The third and final step is the mixing and homogenization of the emulsion and gel [8,104].

## 4. Types of Emulgel

There are two types of emulgels based on the type of emulsion included in their composition, namely oil-in-water (O/W) and water-in-oil (W/O) emulgels. Both types are widely used in the pharmaceutical industry as vehicles for delivering drugs into the skin [8]. Oil-in-water emulgels are used to deliver lipophilic drugs and they are more common, while the W/O type is used to deliver hydrophilic drugs [104].

Depending on the particle size of droplets of emulsion and their distribution, emulgel can be a macroemulsion gel, nanoemulgel, and microemulsion gel (Figure 4) [83].

### 4.1. Macroemulsion Gel

Macroemulsion gels are the most common type of emulgels. The particle size of droplets is usually greater than 400 nm, which leads to the opacity of emulgel and the individual droplets can be easily seen under an optical microscope. Macroemulsions, like all emulsions, are thermodynamically unstable systems but can be stabilized using surface-active agents [6].

### 4.2. Nanoemulgel

When a nanoemulsion is incorporated into a gel, nanoemulgels ensue. Nanoemulsion is among the most successful delivery systems for lipophilic and low bioavailable drugs compared to other nanolipidal delivery systems. Nanoemulsions are optically either transparent or translucent isotropic systems. Nanoemulsions have several merits, such as enhanced physical stability, a high drug-loading capacity of both lipophilic and hydrophilic drugs, and better solubility enhancement capacity. Nanoemulsion formulations possess better transdermal and dermal delivery properties both in vitro and in vivo compared to conventional topical formulations [6,105]. The particle size of nanoemulsions is commonly in the range of 100 to 400 nm [106]. Due to the small size of the particles, nanoemulsions have a greater surface area, which enables better absorption. The nano size of the particles increases the efficiency of any drugs to cross biological membranes, such as the skin [107]. Algahtani et al. have formulated a nanoemulgel with thymoquinone that showed better penetration through the skin and deposition characteristics application compared to conventional hydrogel after topical administration [108]. Furthermore, another study confirmed that nanoemulgel showed enhanced drug permeation and increased transdermal flux as compared to either conventional emulgel or gel [50].

### 4.3. Microemulsion Gel

Microemulsions are usually defined as a thermodynamically stable, transparent, and isotropic system with a droplet size usually in the range of 10–100 nm and are made up of water, oil, and a surfactant, mostly in association with a cosurfactant [109,110,111].

Microemulsions possess physical properties such as transparency, low viscosity, and small particle size and have the ability to form spontaneously in contrast to conventional emulsions. In addition, microemulsions do not show phase separation over a wide temperature range [11].

Microemulsions differ significantly from nano- and macroemulsions. The first difference is in the size of the particles, then the formation and stability. Nanoemulsions require a lower percentage of the emulsifier for preparation compared to microemulsions. Microemulsions are the most stable and can remain stable for several years [112].

It should be noted that it is not only the droplet size that determines if a certain system is thermodynamically stable or unstable. In general, microemulsions consist of high concentrations of emulsifiers (more than 10%), often accompanied by cosurfactants as structurally complementary components, such as amphiphilic short-chain alcohols or their esters. The molecular structure of the surfactants and cosurfactants, as well as their concentrations, determine the system microstructure. If properly chosen, they form a highly integrated layer at the oil-water interface. Therefore, microemulsions form spontaneously and they are characterized by small uniform-sized droplets [11].

A disadvantage of microemulsion is its limited use due to its low viscosity, which leads to a reduction in the contact of the product with the skin. To overcome this disadvantage, viscosity-increasing agents are added to the microemulsion system to form a gel microemulsion, providing appropriate viscosity and prolonged release of the active ingredient [113].

## 5. Advantages of Emulgel Formulations

There are numerous advantages of using emulgels, and therefore the need for emulgels in the pharmaceutical and food industries is increasing.

The first and basic advantage of emulgels is the possibility of incorporating hydrophobic drugs by their incorporation in the oil phase of the emulsion. The prepared emulsion can be easily mixed with the gel base. In this way, the problem of low solubility of hydrophobic drugs can be overcome and their delivery and penetration through the skin can be achieved [5,114].

Ajazuddin et al. have emphasized the advantage of emulgels in releasing the drug compared to frequently used topical preparations such as ointments, creams, lotions, and pastes, which contain a large number of oil excipients that provide emollient properties but retard the drug release. Such hydrophobic excipients do not allow the inclusion of water and the aqueous phase in the formulation, while gel-based formulations provide an aqueous environment, making the preparation less thick and less greasy. Consequently, an emulsion-based gel is a suitable medium for the release and delivery of hydrophobic active substances (drug, cosmetic, and food ingredients) [60].

Controlled release and targeted drug delivery is provided by emulgel through the dual controlled release mechanisms, due to the presence of an emulsion and a gel system. Consequently, emulgel can prolong the effect of drugs by controlling the release of the drug from the formulation so may allow a longer period of drug action [115].

Emulgels have properties such as good spreadability, being greaseless, thixotropy, and good shelf-life, and are pleasant compared to ointments and creams which possess less spreadability, stickiness, and a need to apply through rubbing. Additionally, emulgels are easy to apply on the surface of the skin and possess several advantages since they are easily removable but emollient as well, so they enhance patient compliance [8,116].

Emulgels, when properly formulated, generally have good physical stability with low interfacial tension that is achieved by adding a suitable emulsifier, which consequently leads to a longer shelf life. Other transdermal formulations such as creams and ointment are generally less stable than emulgels. Creams may show phase inversion or breaking and ointments may show rancidity due to their oily base [83,117].

Emulgels show better loading capacity than some other novel drug delivery systems such as niosomes and liposomes which may result in leakage and lesser entrapment efficiency due to their vesicular nature. However, emulgels have a network that can show comparatively better loading capacity and entrapment of active substances [8,118]. Preparation of vesicular drug delivery systems such as niosomes and liposomes needs intensive sonication which may lead to drug degradation and leakage. However, no sonication is required during emulgel production, so this problem is solved [114].

The last and no less important advantage of emulgels is production feasibility and low preparation cost that comprise simple and short processing steps. No highly specialized instruments are needed for the production of emulgel. In addition, materials that are used for the preparation of emulgels are usually easily available and low-cost [119].

## 6. Disadvantages of Emulgel Formulations

In addition to the numerous benefits of emulgels, they also have some disadvantages. One of the disadvantages is the entrapment of bubbles during the preparation process [120] but this can be overcome by sonicating the formed gel for 15 min [58]. Further, emulgel can lead to skin irritation in people with contact dermatitis and can cause allergic reactions [120]. Generally, the disadvantage of topical drug delivery systems is the poor permeability of the drug through the skin due to the thick and complex skin structure, so large drug particles are hardly absorbed through the skin. However, this issue can be partly solved by the addition of penetration enhancers in the emulgel formulation. Despite its benefits, emulgel is a good choice as a delivery system for hydrophobic drug substances only [6].

## 7. Conclusions

Emulgels are novel, modern drug delivery systems that possess favorable features of both emulsion and gels and therefore they have attracted the attention of scientists in the last decade. Emulsion gels are proven effective and convenient delivery systems for various bioactive substances due to their tunable attractive properties and unique characteristics. The advantages of emulgels make them promising formulations for targeted drug delivery with controlled release, which are able to encapsulate and protect various drug, cosmetic, and food ingredients applied topically or perorally. The use of proteins and other ingredients of natural origin in emulgel formulations is currently on the rise due to public health concerns affecting the more frequent replacement of synthetic ingredients with natural ones. The use of emulgels is steadily increasing in the pharmaceutical and cosmetic industry, as well as in the food industry, with great potential to advance further in the future.

## Figures and Tables

**Figure 1 polymers-15-02302-f001:**
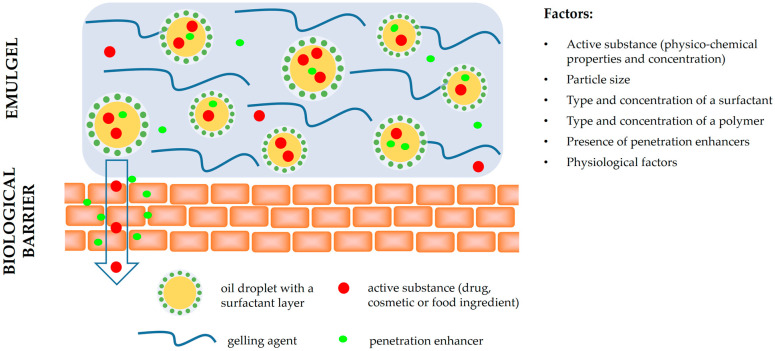
Factors affecting the release of active substances from emulgel formulation and their permeation.

**Figure 2 polymers-15-02302-f002:**
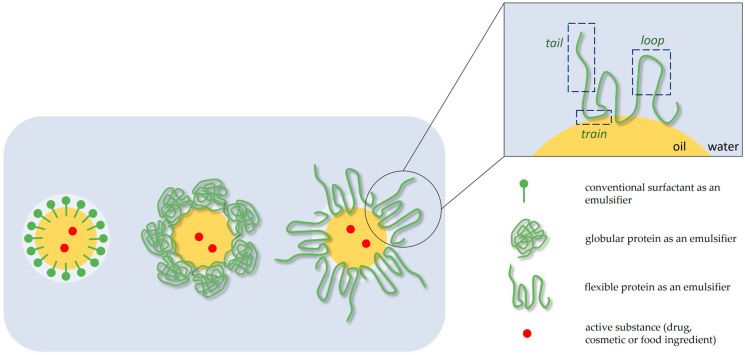
Configuration of conventional and polymeric emulsifiers at oil/water interfaces and the ‘train-loop-tail’ model of adsorption of flexible proteins.

**Figure 3 polymers-15-02302-f003:**
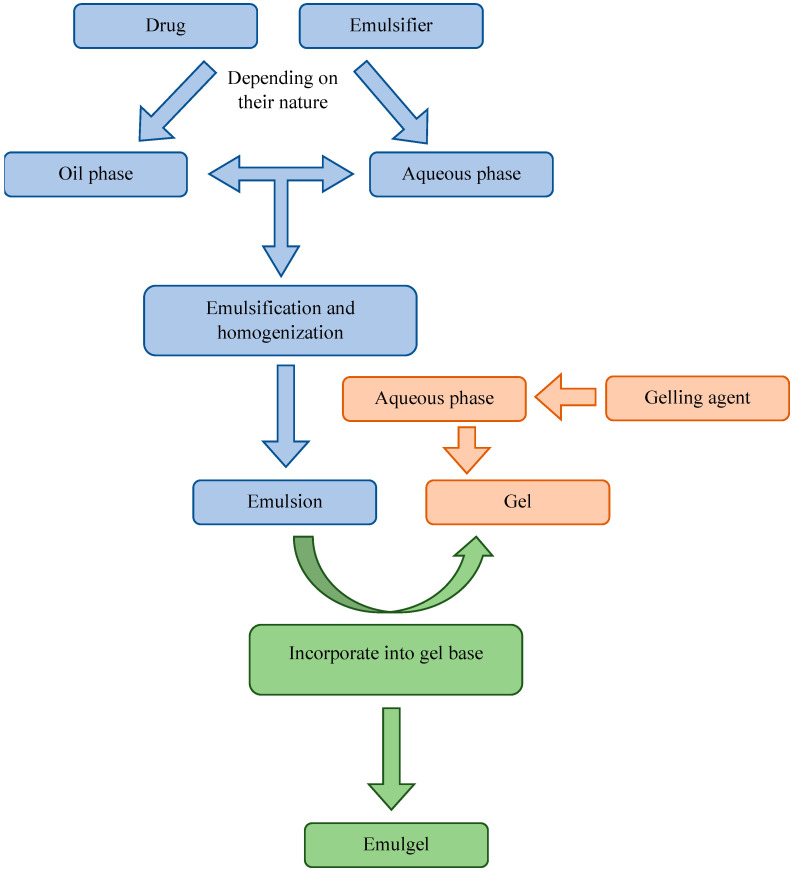
Steps in the process of emulgel preparation.

**Figure 4 polymers-15-02302-f004:**
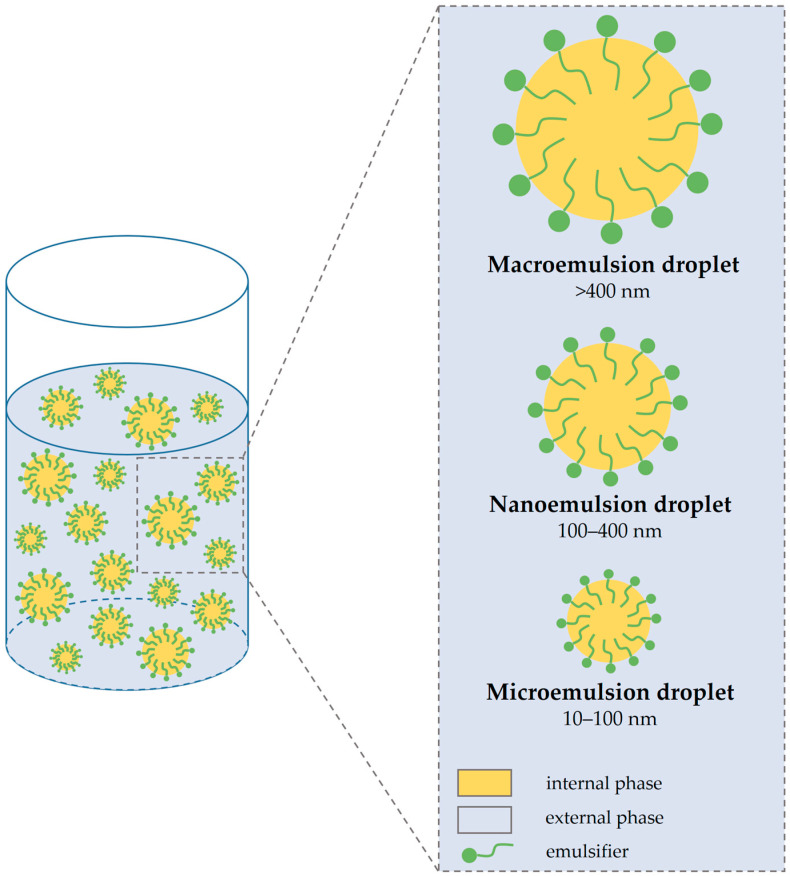
Types and typical droplet diameter sizes of macro-, nano-, and microemulsions.

**Table 1 polymers-15-02302-t001:** List of various gelling agents used in emulgel formulations.

Gelling Agents	Concentration Used (%w/w)	Type	Reference
Carbopol 934	0.5–2.0	synthetic	[41]
Carbopol 940	0.75–2.0	synthetic	[42]
	1.0		[43]
Carbopol 980	0.5	synthetic	[44]
	0.7		[45]
Poloxamer 407	12.5	synthetic	[46]
Sepineo-P600	6.0	synthetic	[47]
HPMC	4.0–6.0	semi-synthetic	[48]
HEC *	2.5	semi-synthetic	[49]
NaCMC	1.0	semi-synthetic	[50]
Guar gum	0.5	natural	[51]
Xanthan gum	0.75 and 1.0	natural	[52]
Whey protein isolate	5.0	natural	[53]
Soybean protein isolate	10.0	natural	[54]
Alginate	2.0	natural	[38]
Combination of Carbopol 934 and 940	0.5–2.0 and 1.0–2.0	synthetic	[55]
Combination of Carbopol 934 or 940 and HPMC	1.0 and 0.5–1	synthetic and semi-synthetic	[56]
Combination of Carbomer interpolymer type A and Xanthan	0.45 and 0.20	synthetic and natural	[57]
Combination of Xanthan gum and Chitosan	1.5 and 2.0	natural	[58]

* HEC—hydroxyethyl cellulose.

**Table 2 polymers-15-02302-t002:** List of emulsifiers commonly used in emulgel formulations.

Emulsifiers	Concentration (%w/w)	Type	Reference
Tween 80	1.0	nonionic	[83]
Combination of Tween 80 and Span 80	0.30 or 0.50 and 0.45 or 0.75	nonionic	[84]
	1.0 and 1.5		[85]
	2.0 and 3.0		
Combination of Tween 60 and Span 60	0.5 and 4.50	nonionic	[86]
Combination of Tween 20 and Span 20	0.5 and 1.0	nonionic	[87]
	0.4 and 0.3		[88]
	1.0 and 3.0		[64]
Soybean protein isolate	7.0	amphiphilic	[89]
	4.0		[90]

**Table 4 polymers-15-02302-t004:** List of preservatives commonly used in emulgel formulations.

Preservatives	Concentration Used (%w/w)	Reference
Methyl paraben	0.03	[87]
	0.05	[100]
	0.15	[64]
Combination of methyl paraben and propyl paraben	0.03 and 0.01	[101]
	0.03 and 0.05	[47]
Phenoxyethanol	0.2	[102]
Benzalkonium chloride	0.01	[103]

## Data Availability

Not applicable.

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
