# Peer review of "Emulgels: Promising Carrier Systems for Food Ingredients and Drugs"

_polymers, 2023, doi:10.3390/polym15102302_

Round 1

Reviewer 1 Report

Below are some specific issues that should be taken into consideration and amended before acceptance for publication.

Line 53: please rephrase or elaborate on “distinguishing quality”

Line 56: “long shelf life [10]”
The shelf-life of a drug product depends on several factors, for instance, its physical instability, and the chemical instability of both the active pharmaceutical ingredients(s) as well as of the excipients. Stating that Emulgels possess a long shelf-life is an unacceptable generalization. 

57-58: “Compared to other topical and transdermal formulations, emulsion-based formulations possess better chemical, physical and biological compatibility with the skin.”
The above sentence is too general and lacks support!

186 – add “physical" in “and to ensure the physical stability of the emulsion during its shelf life”

189 – delete biologically

196 – you mention “by being adsorbed on the oil–water interface leading to reduced interfacial tension and viscoelastic interface film”. That is also what happens to the previously mentioned surfactants (for instance polysorbate, sorbitan monolaurate) – they lay at the interface and act by reducing the interfacial tension. Please rephrase.

194 – delete “outstanding”. “Proteins possess outstanding amphiphilic,”

200 – “Table 2 lists some the emulsifiers”  as this is not an exhaustive list and other surfactants and concentrations may be successfully used as well.

205 – replace “ameliorate” by “increase” or another clear qualifier.

227 – define TEA when first using (Is it triethanolamine ?)

231 – “2.7 Preservatives” are these effective against fungi, bacteria, or both? Are they usually used isolated or in association?

Substances in this class (preservatives) are not exclusive to emulgels and are used in most liquid and semisolid drug products and in food products as well. Therefore additional / broader information based on their use should be included as well as mention to some particular with emulgels - for instance, are there some known incompatibility between preservatives and the gelling substances?

241- “…improve the quality of the formulation…” Quality has a precise meaning that is not appropriate in the sentence. Please replace it with a suitable word, for instance, "characteristics", "attributes" or another.

248 – “This step includes heating of the phases and incorporating of the drug into emulsion as required.”

Please clarify where is the drug added to? Dissolved in the internal phase prior to emulsification? To the prepared emulsion? Both?

252 - Fig. 1 should also depict the incorporation of the drug substance

265 – “seen under a microscope.” Replace with “seen under an optical microscope.”

265 – “Macroemulsions are thermodynamically unstable systems, but can be stabilized using surface-active agents”. All disperse systems, including emulsions are thermodynamically unstable, not only macroemulsions! Please rephrase or elaborate.

270 – please provide support for the sentence “Nanoemulsions are thermodynamically stable”

272 – “Nanoemulsions have several merits like enhanced stability,”, what kind of stability are the authors referring to? Chemical, physical, both?

285 – Microemulsions, include kinetically stable but thermodynamically unstable emulsions, as addressed in Ref 11. This utmost important aspect and differentiation between thermodynamically stable vs unstable dispersions should be discussed. It is not a question of particle size as apparently, the authors state.

301 – Fig 2. oil phase and aqueous should be replaced by internal and external phase, respectively as  the illustration applies to both O/W and W/O emulsions. Please elaborate on this “hard-limits“ classification (10-100 nm, 100-400 nm, >400nm). Are these official limits? Who proposed such limits? Are they flexible?

321 – “…emulgel can prolong the effect of drugs with short half-life [88]”.

The nature of the carrier, emulgel in this particular case, does not change the effect of the drug substance nor its distribution or elimination kinetics.  As such, the sentence is misleading and should be rephrased to convey the idea that the emulgel (due to slowing down the drug substance release, and consequently the absorption of the drug substance) may allow a longer time period of action for the drug product (controlled, extended release).

322 – Some of these properties or attributes are common to other topical formulations and as such are not exclusive to emulgels. Please rephrase.

327 – “Emulgels, when properly formulated, have generally good thermodynamic stability with low interfacial tension that is made by adding a emulsifier to improve shelf-life stability”…

The longer shelf-life is a consequence of the enhanced physical stability that is achieved by adding suitable surfactants/emulsifiers. The sentence should be rephrased.

 329 – The sentences “Other transdermal formulations such as creams and ointment are comparatively less stable than emulgels. Creams show phase inversion or breaking and ointments show rancidity due to oily base [60]” are generalizations and are not acceptable in the current form. The use of “generally less stable” or “may show phase inversion…” or “may show rancidity due to oily base..” is an acceptable approach in my opinion.

335 – “Preparation of vesicular molecules needs…” please clarify the meaning of “vesicular molecules”

340 – “No specialized instruments are needed for the production of emulgel.” Please clarify what are “specialized instruments” or rephrase.

345- “One of the disadvantages is the entrapment of bubbles during the preparation process.”

This is an issue not exclusive to emulgels and can be easily overcome by performing the emulsification and or mixing steps under reduced pressure (vacuum) without using expensive equipment. Please rephrase.

The quality of the English writing is acceptable, in my opinion, but some minor style adjustments should be made. 

For instance, the authors use frequently the word “agent” when referring to a (chemical) substance (active or not), for instance, “Consequently, an emulsion-based gel is a suitable medium for release and delivery of hydrophobic active agents (drug, cosmetic and food ingredients)”. It is my opinion that the general use of “agent” is not acceptable and sentences should be rephrased. Example:  “Consequently, an emulsion-based gel is a suitable medium for the release and delivery of hydrophobic substances (drug, cosmetic and food ingredients)” or "drugs".

Author Response

Dear Reviewer,

Firstly, on behalf of all co-authors, I would like to thank you for helping us with all your comments and suggestions to improve our initial version of the manuscript. We accepted all suggestions, and they are included in this modified and improved version of our manuscript.

We have highlighted these changes in the manuscript.

All grammatical, spelling and other language-related errors have been corrected by a native English speaker.

In this letter we tried to state point by point the changes we made in the manuscript according to your comments.

Our team hopes that we have managed to properly consider and act upon the comments and suggestions raised.

Yours sincerely,

Veljko Krstonošić, corresponding author

Reviewer #1

  1. Reviewer’s comment:

Line 53: please rephrase or elaborate on “distinguishing quality”

Our action:

We rephrased “distinguishing quality” as “numerous preferable characteristics”.

  1. Reviewer’s comment:

Line 56: “long shelf life [10]”

The shelf-life of a drug product depends on several factors, for instance, its physical instability, and the chemical instability of both the active pharmaceutical ingredients(s) as well as of the excipients. Stating that Emulgels possess a long shelf-life is an unacceptable generalization.

Our action:

We completely agree and we deleted “long shelf life” to avoid the misleading generalization.

  1. Reviewer’s comment:

57-58: “Compared to other topical and transdermal formulations, emulsion-based formulations possess better chemical, physical and biological compatibility with the skin.”

The above sentence is too general and lacks support!

Our action:

Thank you very much for your suggestion, we explained in detail compared to which topical and transdermal formulations, emulgels possess better compatibility with the skin.

  1. Reviewer’s comment:

186 – add “physical" in “and to ensure the physical stability of the emulsion during its shelf life”

Our action:

We added “physical” to the appropriate place.

  1. Reviewer’s comment:

189 – delete biologically

Our action:

We deleted “biologically”.

  1. Reviewer’s comment:

196 – you mention “by being adsorbed on the oil–water interface leading to reduced interfacial tension and viscoelastic interface film”. That is also what happens to the previously mentioned surfactants (for instance polysorbate, sorbitan monolaurate) – they lay at the interface and act by reducing the interfacial tension. Please rephrase.

Our action:

We agree with your suggestion that this also happens with previously mentioned conventional surfactants, we thus rephrased that part and added some specific properties of proteins as emulsifiers in this section.

  1. Reviewer’s comment:

194 – delete “outstanding”. “Proteins possess outstanding amphiphilic,”

Our action:

We deleted “outstanding”.

  1. Reviewer’s comment:

200 – “Table 2 lists some the emulsifiers”  as this is not an exhaustive list and other surfactants and concentrations may be successfully used as well.

Our action:

Thank you very much for your comment, we completely agree that this is not an exhaustive list like the other lists and we made a correction accordingly.

  1. Reviewer’s comment:

205 – replace “ameliorate” by “increase” or another clear qualifier.

Our action:

We replaced the term “ameliorate” with appropriate term “increase”.

  1. Reviewer’s comment:

227 – define TEA when first using (Is it triethanolamine ?)

Our action:

Thank you for your comment, we corrected this omission and defined TEA (triethanolamine).

Our action:

  1. Reviewer’s comment:

231 – “2.7 Preservatives” are these effective against fungi, bacteria, or both? Are they usually used isolated or in association?

Substances in this class (preservatives) are not exclusive to emulgels and are used in most liquid and semisolid drug products and in food products as well. Therefore additional / broader information based on their use should be included as well as mention to some particular with emulgels - for instance, are there some known incompatibility between preservatives and the gelling substances?

Our action:

The ideal preservative should be active against most microorganisms (both bacetria and fungi) at a low concentration but this is not always the case. In literature we found that preservatives are used both isolated and associated, but combining two preservatives can increase the spectrum of action. We have stated examples of the specific effectiveness of some preservatives and have added some specifics related to preservatives in emulgel formulations such as interactions and incompatibilities.

  1. Reviewer’s comment:

241- “…improve the quality of the formulation…” Quality has a precise meaning that is not appropriate in the sentence. Please replace it with a suitable word, for instance, "characteristics", "attributes" or another.

Our action:

We appreciate your suggestion and we replaced “quality” with a more suitable term.

  1. Reviewer’s comment:

248 – “This step includes heating of the phases and incorporating of the drug into emulsion as required.”

Please clarify where is the drug added to? Dissolved in the internal phase prior to emulsification? To the prepared emulsion? Both?

Our action:

Thank you for your suggestion, we clarified where the drug substance is added. The drug usually added into internal phase of emulsion, which is utilized as a carrier for drug substances.

  1. Reviewer’s comment:

252 - Fig. 1 should also depict the incorporation of the drug substance

Our action:

We modified Figure 1 so we depicted the step of incorporating the drug into the formulation.

  1. Reviewer’s comment:

265 – “seen under a microscope.” Replace with “seen under an optical microscope.”

Our action:

We added your suggestion in our manuscript.

  1. Reviewer’s comment:

265 – “Macroemulsions are thermodynamically unstable systems, but can be stabilized using surface-active agents”. All disperse systems, including emulsions are thermodynamically unstable, not only macroemulsions! Please rephrase or elaborate.

Our action:

Thank you very much for pointing out the omission, we agree and we rephrased the sentence.

  1. Reviewer’s comment:

270 – please provide support for the sentence “Nanoemulsions are thermodynamically stable”

Our action:

Due to previously indicated mistakes we deleted this statement.

  1. Reviewer’s comment:

272 – “Nanoemulsions have several merits like enhanced stability,”, what kind of stability are the authors referring to? Chemical, physical, both?

Our action:

We referred to physical stability and we rephrase this sentence.

  1. Reviewer’s comment:

285 – Microemulsions, include kinetically stable but thermodynamically unstable emulsions, as addressed in Ref 11. This utmost important aspect and differentiation between thermodynamically stable vs unstable dispersions should be discussed. It is not a question of particle size as apparently, the authors state.

Our action:

Thank you for the valuable comment and suggestion. We modified this part of the manuscript in order to emphasize that it is not only the droplet size that determines if a certain system is thermodynamically stable or unstable. The specific microemulsion composition (such as high levels of emulsifiers, often accompanied with cosurfactants), in this respect, was additionally discussed in the text.

  1. Reviewer’s comment:

301 – Fig 2. oil phase and aqueous should be replaced by internal and external phase, respectively as  the illustration applies to both O/W and W/O emulsions. Please elaborate on this “hard-limits“ classification (10-100 nm, 100-400 nm, >400nm). Are these official limits? Who proposed such limits? Are they flexible?

Our action:

Thank you for your comments, we corrected Figure 2 and agree with you that is "hard limits" due to in literature can be found different limits and they are flexible. Consequently, we corrected and emphasized that these are the usual limits.

  1. Reviewer’s comment:

321 – “…emulgel can prolong the effect of drugs with short half-life [88]”.

The nature of the carrier, emulgel in this particular case, does not change the effect of the drug substance nor its distribution or elimination kinetics.  As such, the sentence is misleading and should be rephrased to convey the idea that the emulgel (due to slowing down the drug substance release, and consequently the absorption of the drug substance) may allow a longer time period of action for the drug product (controlled, extended release).

Our action:

Thank you very much for your comment, we rephrased the sentences to avoid misleading.

  1. Reviewer’s comment:

322 – Some of these properties or attributes are common to other topical formulations and as such are not exclusive to emulgels. Please rephrase.

Our action:

We agree that some topical formulations possess some of these properties, but no formulations possess all of these advantages. We have improved this section with a comparison of the properties of emulgels and other topical formulations

  1. Reviewer’s comment:

327 – “Emulgels, when properly formulated, have generally good thermodynamic stability with low interfacial tension that is made by adding a emulsifier to improve shelf-life stability”…

The longer shelf-life is a consequence of the enhanced physical stability that is achieved by adding suitable surfactants/emulsifiers. The sentence should be rephrased.

Our action:

We appreciate of your suggestion and we modified the sentence.

  1. Reviewer’s comment:

329 – The sentences “Other transdermal formulations such as creams and ointment are comparatively less stable than emulgels. Creams show phase inversion or breaking and ointments show rancidity due to oily base [60]” are generalizations and are not acceptable in the current form. The use of “generally less stable” or “may show phase inversion…” or “may show rancidity due to oily base..” is an acceptable approach in my opinion.

Our action:

We accepted all your suggestions and we modified those sentences.

  1. Reviewer’s comment:

335 – “Preparation of vesicular molecules needs…” please clarify the meaning of “vesicular molecules”

Our action:

Vesicular molecules are referred of vesicular drug delivery systems like liposomes and niosomes and we rephrased this term to avoid misleading.

  1. Reviewer’s comment:

340 – “No specialized instruments are needed for the production of emulgel.” Please clarify what are “specialized instruments” or rephrase.

Our action:

In this case we think of specialized instruments such as extruder, membrane contractor, high-pressure eductor nozzle-based which are needed to prepare other delivery systems like liposomes in compared to emulgels where are not needed.

  1. Reviewer’s comment:

345- “One of the disadvantages is the entrapment of bubbles during the preparation process.”

This is an issue not exclusive to emulgels and can be easily overcome by performing the emulsification and or mixing steps under reduced pressure (vacuum) without using expensive equipment. Please rephrase.

Our action:

We agree that is not exclusive to emulgels and it is mainly a disadvantage of gels in general, so we rephrased that. In literature we found that researchers remove entrapped air bubbles by sonicating gels for 15 minutes as well (in addition to mixing under reduced pressure).

* Additional Reviewer’s comment:

The quality of the English writing is acceptable, in my opinion, but some minor style adjustments should be made.

For instance, the authors use frequently the word “agent” when referring to a (chemical) substance (active or not), for instance, “Consequently, an emulsion-based gel is a suitable medium for release and delivery of hydrophobic active agents (drug, cosmetic and food ingredients)”. It is my opinion that the general use of “agent” is not acceptable and sentences should be rephrased. Example:  “Consequently, an emulsion-based gel is a suitable medium for the release and delivery of hydrophobic substances (drug, cosmetic and food ingredients)” or "drugs".

Our action:

Thank you for this suggestion, we replaced the term “agent” in the text. Besides, other grammatical, spelling and other language-related errors have been corrected by a native English speaker.

Reviewer 2 Report

This review summarized recent advances in emulgels as the carrier systems for food ingredient and drugs. The authors present the colloids from several different aspects e.g. formulation of emulgel, types of emulgel, and advantages/disadvantages of emulgel formulation.

The content is more detailed, but there are almost no pictures of the research content. This review has only 2 figures and cannot show in detail the progress in the related fields. It is recommended that the author add. 

The cited references are mostly recent publication and relevant. The statements and conclusions are drawn coherent and supported by the listed citations.

However, there were similar review published recently such as:
- Drug Development and Industrial Pharmacy 47.8 (2021): 1193-1199.
- Journal of Young Pharmacists 13(1) (2021): 76.

Overall, the content of this article is immature and does not meet the standards for journal publication

Author Response

Dear Reviewer,

Firstly, on behalf of all co-authors, I would like to thank you for helping us with all your comments and suggestions to improve our initial version of the manuscript. We accepted your suggestions, and they are included in this modified and improved version of our manuscript.

We have highlighted these changes in the manuscript.

All grammatical, spelling and other language-related errors have been corrected by a native English speaker.

Our team hopes that we have managed to properly consider and act upon the comments and suggestions raised.

Yours sincerely,

Veljko Krstonošić, corresponding author

Reviewer #2

Reviewer’s comment:

This review summarized recent advances in emulgels as the carrier systems for food ingredient and drugs. The authors present the colloids from several different aspects e.g. formulation of emulgel, types of emulgel, and advantages/disadvantages of emulgel formulation.

The content is more detailed, but there are almost no pictures of the research content. This review has only 2 figures and cannot show in detail the progress in the related fields. It is recommended that the author add.

The cited references are mostly recent publication and relevant. The statements and conclusions are drawn coherent and supported by the listed citations.

However, there were similar review published recently such as:

- Drug Development and Industrial Pharmacy 47.8 (2021): 1193-1199.

- Journal of Young Pharmacists 13(1) (2021): 76.

Overall, the content of this article is immature and does not meet the standards for journal publication

Our action:

Firstly, we would like to thank you for the positive parts of your review regarding the use of recent and relevant literature data in a detailed manner and the coherent conclusions of our review article. Besides, we cordially appreciate your critical comments and the time you invested to review our manuscript.

In order to improve our initial version of the manuscript, we substantially modified the text. We accepted your suggestion and added one more figure summarizing and illustrating the factors affecting the release of active substances from emulgels and their permeation through the membranes, which is necessary for a formulation to exert its biological/pharmacological effect. To the best of our knowledge, there are no similar figures in the literature, describing all these factors in one place and in an illustrative manner.

We agree that there are recent publications on the topic of emulgels, including the two that you stated. However, there are large and significant differences between those articles and ours. The study of Talat et al. (Drug Dev Ind Pharm) discusses only topical drug delivery systems and it contains one table and one figure. The text is also less comprehensive about formulation composition and the influence of formulation factors on drug release and skin penetration (only physiological factors and physicochemical factors of drug substances are discussed), and we consider of utmost importance to explain in detail and discuss the impact of excipients on the aforementioned processes. The paper of Charyulu et al. (J Young Pharm) is published as a ‘short communication’, which brings very interesting brief overview of topical emulgels (with 15 references) to the readers who are not very familiar with this topic (serving as an excellent starting point). Nevertheless, we highly appreciate the work of all authors in this field, and both these articles were already cited in the initial version of our manuscript.

As the main novelty of our work, we would point out that we discussed the use of emulgels as carriers not only for drug substances, but also for food ingredients, which is not the case for the previously mentioned two papers that are only focused on topical use of emulgels. In order to emphasize the significance of emulgel formulations in peroral delivery, we added and summarized the most recent advances in this field (as added in ‘Introduction’ and parts about gelling agents and emulsifiers, with 15 new references and more than 1000 words). In these parts, the main benefits of the use of emulgels in food and nutraceutical industry have been discussed, as well as the most recent innovations.

Eventually, according to the suggestions of another reviewer, we clarified several important points, we emphasized the differences between macro-, nano- and micromeulsion gels, we added some specific properties of proteins as emulsifiers in emulgel formulations, and we made some other corrections and modifications.

After all the changes we made in our manuscript based on your comments and the comments of another reviewer, we hope that the manuscript is much approved now and that would be suitable for publication.

Round 2

Reviewer 2 Report

1)The author has improved and revised this version compared to the previous one, but the content of the pictures is still insufficient, and more figures are needed as an overview.

2)Some of the cited images need to be marked from where the cited image.

3)Comparison with other systems is necessary, such as polypeptides, SiO2, polymeric hollow spheres, etc.

4) Some related research about the drug delivery polymers should be cited to highlight the potential applications of these materials. Biomacromolecules 2021, 22, 2, 732–742; Pharmaceutics, 2023, 15(2): 368.

Author Response

Dear Reviewer,

On behalf of all co-authors, I would like to thank you for your comments that we fully accepted and that helped us to additionally improve our manuscript.

In this letter we tried to state point by point the changes we made in the manuscript according to your comments. We have highlighted these changes in the manuscript as well.

Reviewer’s comments:

1)The author has improved and revised this version compared to the previous one, but the content of the pictures is still insufficient, and more figures are needed as an overview.

Answer:

Thank you for your positive comments and a suggestion. We accepted it and added one more figure about the role of different types of emulsifiers (conventional and macromolecular) in emulgel stabilization. The part ‘2.4. Emulsifiers’ has been additionally modified accordingly. Now there are four figures and four tables in the text, which contributes to the comprehensiveness of the manuscript.

2)Some of the cited images need to be marked from where the cited image.

Answer:

All the images/figures have been created by the authors and they represent the original work.

3)Comparison with other systems is necessary, such as polypeptides, SiO2, polymeric hollow spheres, etc.

Answer:

There are large differences between emulgels as drug delivery systems and the aforementioned systems. We added the comparative analysis of their structures and properties in the manuscript. It was added in the manuscript after the brief introduction of emulgel formulations and the influence of formulation factors on drug release and transport (part ‘2. Formulation of emulgel’). Besides, the results on Pickering emulsions stabilized by SiO2 particles and amphoteric lignin have been added in the part ‘2.4. Emulsifiers’.

4) Some related research about the drug delivery polymers should be cited to highlight the potential applications of these materials. Biomacromolecules 2021, 22, 2, 732–742; Pharmaceutics, 2023, 15(2): 368.

Answer:

In accordance with the previous comment and the comparative analysis of emulgels and other polymeric drug delivery systems, we added several new references, including the two suggested studies.

*************

Our team hopes that we have managed to properly consider and act upon the comments and suggestions raised.

Yours sincerely,

Veljko Krstonošić, corresponding author

Round 3

Reviewer 2 Report

 Accept in present form